# Area-Detector Computed Tomography for Pulmonary Functional Imaging

**DOI:** 10.3390/diagnostics13152518

**Published:** 2023-07-28

**Authors:** Yoshiharu Ohno, Yoshiyuki Ozawa, Hiroyuki Nagata, Shuji Bando, Shang Cong, Tomoki Takahashi, Yuka Oshima, Nayu Hamabuchi, Takahiro Matsuyama, Takahiro Ueda, Takeshi Yoshikawa, Daisuke Takenaka, Hiroshi Toyama

**Affiliations:** 1Department of Diagnostic Radiology, Fujita Health University School of Medicine, Toyoake 470-1192, Aichi, Japan; 2Joint Research Laboratory of Advanced Medical Imaging, Fujita Health University School of Medicine, Toyoake 470-1192, Aichi, Japan; nagata_zaiken@yahoo.co.jp; 3Department of Radiology, Fujita Health University School of Medicine, Toyoake 470-1192, Aichi, Japan; yoshiyuki.ozawa@fujita-hu.ac.jp (Y.O.);; 4Department of Diagnostic Radiology, Hyogo Cancer Center, Akashi 673-0021, Hyogo, Japan

**Keywords:** lung, multidetector computed tomography, area-detector CT, pulmonary function, radiation dose

## Abstract

An area-detector CT (ADCT) has a 320-detector row and can obtain isotropic volume data without helical scanning within an area of nearly 160 mm. The actual-perfusion CT data within this area can, thus, be obtained by means of continuous dynamic scanning for the qualitative or quantitative evaluation of regional perfusion within nodules, lymph nodes, or tumors. Moreover, this system can obtain CT data with not only helical but also step-and-shoot or wide-volume scanning for body CT imaging. ADCT also has the potential to use dual-energy CT and subtraction CT to enable contrast-enhanced visualization by means of not only iodine but also xenon or krypton for functional evaluations. Therefore, systems using ADCT may be able to function as a pulmonary functional imaging tool. This review is intended to help the reader understand, with study results published during the last a few decades, the basic or clinical evidence about (1) newly applied reconstruction methods for radiation dose reduction for functional ADCT, (2) morphology-based pulmonary functional imaging, (3) pulmonary perfusion evaluation, (4) ventilation assessment, and (5) biomechanical evaluation.

## 1. Introduction

Since the clinical installation of a multidetector row CT (MDCT) in 1999 [1], detector rows have been increased from 4 to 64 by every vendor almost every 2 years to result in area-detector CT (ADCT) systems with 256- or 320-detector rows, which are now widely used in routine clinical practice worldwide since their introduction in 2007. With ADCT, isotropic volume data for the entire brain, heart, or some other organs, as well as for entire tumors, can be acquired almost immediately within an area of nearly 160 mm without helical scanning. Whole-organ-perfusion CT data can, thus, be obtained by means of continuous dynamic scanning, allowing for the qualitative and/or quantitative evaluation of the perfusion of some organs as well as of pulmonary nodules, lymph nodes, or lung cancer for a variety of clinical aims [2,3,4,5,6,7,8,9,10,11,12,13,14,15,16,17,18,19,20,21,22,23]. Moreover, the system can obtain CT data with both helical scanning and step-and-shoot or wide-volume scanning for body CT imaging. In addition, ADCT has the potential to perform dual-energy CT and subtraction CT, which is another promising technique for enhancing the visualization of contrast enhancement by using iodine contrast media as well as xenon or krypton for functional evaluations. Therefore, ADCT systems are now being used for not only morphological examinations but also functional assessments for a variety of clinical purposes including pulmonary functional imaging.

This review will focus on (1) new reconstruction methods being used for radiation dose reduction for functional ADCT, (2) morphology-based pulmonary functional imaging, (3) pulmonary perfusion evaluation, (4) ventilation assessment, and (5) biomechanical evaluation using ADCT because these items have proven to yield fruitful results during the last few decades.

## 2. New Reconstruction Methods Used for Radiation Dose Reduction for Functional ADCT

For the last decade or so, dose reduction strategies have been relying on a variety of techniques for data acquisition, such as tube current or tube voltage reduction, increased helical pitch, scan length optimization, and the utilization of automatic exposure control (AEC) [24,25,26,27,28,29,30]. In general, the image noise is inversely proportional to the square root of the radiation dose, so reduced-dose CT images have a higher noise level than standard-dose images, and care must be exercised to ensure that the former remain suitable for diagnosis. To overcome the increase in image noise on reduced-dose CT images, various imaging filters, reconstruction algorithms, and kernels have been developed. Since the early 2010s, most CT vendors have clinically installed hybrid-type or model-based iterative reconstruction (IR) algorithms for use in routine clinical practice, and many practitioners have found them useful for cardiothoracic imaging. For hybrid-type IR, hybrid algorithms combine both analytical and iterative methods so that the initial image can be generated by means of analytical methods, followed by the use of iterative methods to reduce noise in the image domain. However, iterative algorithms can also be directly employed for the reconstruction process. The term hybrid IR usually refers to algorithms that mainly reduce image noise through cyclic image processing [27,28,29,30], while the term model-based IR usually refers to algorithms that employ models of the acquisition process, image statistics, and system geometry. Although the clinical performance of IR algorithms is not necessarily related to the complexity of the method [31], the reconstruction of hybrid-type IR algorithms is generally faster than that of model-based IR algorithms, and it is more easily applied in routine clinical practice. In 2019, deep learning reconstruction (DLR) was introduced and clinically installed and tested by a few vendors [32,33,34,35,36,37]. These methods can be grouped into two major categories, indirect and direct DLR frameworks [37]. For indirect DLR frameworks, either filtered back projection (FBP) or IR is used. The three types of indirect frameworks, sinogram-based, image-based, and hybrid, are differentiated based on when the deep learning network is deployed. Sinogram-based frameworks focus on sinogram optimization, and the network is deployed before the sinogram is treated with FBP or IR. For image-based frameworks, the network optimizes the image after initial reconstruction with FBP or IR, while hybrid frameworks combine the sinogram with image optimization [37]. Direct DLR algorithms reconstruct the sinogram directly into an image without the use of FBP or IR. This can reduce artifacts introduced by FBP or IR, but this is only possible if the ground truth images used for model training do not contain FBP- or IR-related artifacts [37]. The currently used hybrid-type and model-based IR and DLR methods from major vendors are shown in Table 1.

During the last few decades, ADCT has been used for the aforementioned reconstruction methods to reduce the radiation dose for various clinical purposes, and the results have been published. One study reported that the image quality obtained with tube currents of 100 mA and 50 mA and an FBP algorithm was significantly lower than that for both protocols using AIDR 3D, one of the hybrid-type IR methods, for image reconstruction [27]. Moreover, all inter-method agreements for bronchiectasis, emphysema, ground-glass opacity, honeycomb pattern, interlobar septal thickening, nodules, and reticular opacity ranged from moderate to substantial or almost perfect. Furthermore, all agreements for the mediastinal and pleural findings among reduced-dose CTs using AIDR 3D algorithms and standard-dose CT using an FBP algorithm were almost perfect [27]. In addition, the Area-Detector Computed Tomography for the Investigation of Thoracic Diseases (ACTIve) study group conducted a multicenter study to assess the image quality and radiation dose reduction in the case of chest CT using AIDR 3D, but their study used standard-dose CT with a tube current of 240 mA and reduced-dose CTs with tube currents of 120 mA and 60 mA [38]. The same group also assessed the utility of the AIDR 3D algorithm for lung nodule detection on reduced-dose CT [39]. No significant differences in solid lung nodule detection were found between reduced-dose CT protocols using tube currents of 120 mA and 20 mA. Moreover, a comparison of ground-glass nodule (GGN) detection capability showed that the capability of the two protocols for detecting GGNs with a diameter equal to or more than 8 mm was not significantly different [39]. Another study found that further radiation dose reduction without significant degradation of subsolid nodule detection was obtained with the same hybrid-type IR method [40]. As a result of these published findings, AIDR 3D for ADCT is currently used in routine clinical practice.

For pulmonary functional imaging, the Quantitative Imaging Biomarkers Alliance (QIBA) of the Radiological Society of North America (RSNA) has been developing QIBA profiles based on in vitro study findings for lung density since 2007 to standardize the CT protocol for 64-detector-row CTs from major vendors [41]. In collaboration with the QIBA, the Japan Quantitative Imaging Biomarker Alliance (J-QIBA) of the Japan Radiological Society has been testing and has confirmed the capabilities of each of the state-of-the-art reconstruction techniques, such as hybrid-type IR, model-based IR, or DLR for the same settings, as well as airway dimension evaluation for not only ADCT but also high-definition CT (HDCT) or ultra-high-resolution CT (UHR-CT) [42,43]. These studies proved that all state-of-the-art reconstruction methods have the potential ability to reduce the radiation dose of chest CT while maintaining the requirements from the QIBA profile for QIBA-recommended phantom studies [42,43]. Therefore, by combining state-of-the-art reconstruction methods with ADCT, pulmonary functional CT can be assessed in routine clinical practice in terms of not only morphological but also functional information by using various procedures detailed in the sections that follow.

## 3. Morphology-Based Pulmonary Functional Imaging

Like other MDCTs, morphology-based pulmonary functional ADCT has been used for the quantitative assessment of chronic obstructive pulmonary disease (COPD), interstitial lung disease (ILD), or other diseases by using not only standard- but also reduced-dose protocols [27,31,38,39,40,41,42,43,44,45,46,47].

For COPD assessments, CT can be used to assess morphological and functional changes related to COPD [48,49,50,51,52,53,54,55,56,57,58,59,60]. During the last few decades, many commercially available and proprietary software products as well as various visual scoring systems have been used for the CT-based assessment of COPD, with two major approaches reported in the literature for quantitative CT assessment of COPD [48,49,50,51,52,53,54,55,56,57,58,59,60]. One is the determination of the percentage of low attenuation area (%LAA) in the lung, which indicates emphysema changes, and the other is the determination of the wall area ratio (WA%) of the bronchi, which indicates bronchial lumen narrowing and bronchial wall thickening [48,49,50,51,52,53,54,55,56,57,58,59,60]. In the past literature [48,49,50,51,52,53,54,55,56,57,58,59,60], it has been suggested that the %LAA has a good correlation with FEV_1_/FVC, %FEV_1_, %DL_CO_, or DL_CO_/V_A_ and that the WA% also has a good correlation with FEV_1_/FVC and %FEV_1_. Therefore, it is suggested that these imaging parameters are useful as quantitative imaging biomarkers for COPD [48,49,50,51,52,53,54,55,56,57,58,59,60]. In addition, three-dimensional (3D) airway luminal volumetry has been introduced as another quantitative method for evaluating the airways of COPD patients [58,59]. Taking the findings provided by these quantitative CT assessments of COPD and the need for radiation dose reduction strategies into consideration [42,43,44,45,46,58,59,60], the application of IR algorithms has been considered an important issue for an accurate quantitative CT evaluation of COPD. One study demonstrated that agreement for the %LAA between standard-dose CT obtained at 300 mA and reduced-dose CT at 50 mA tended to improve when using AIDR 3D rather than conventionally applied FBP [44] (Figure 1).

Moreover, the ACTIve study group applied the same hybrid-type IR algorithm and obtained similar results for standard-dose CT at 240 mA and reduced-dose CTs at 120 mA and 60 mA [60]. Thus, the use of a hybrid-type IR resulted in greater consistency of emphysema quantifications performed on reduced-dose and ultra-low-dose CTs than on standard-dose CT images. Although the %LAA and the WA% have been recommended as the two main quantitative parameters for COPD assessment [41,42,43,44,45,46,47,48,49,50,51,52,53,54,55,56,57,58,59,60], 3D airway luminal volumetry has also been introduced as another method for quantitative ADCT evaluation of airflow limitation in COPD [58,59]. Koyama et al. assessed the utility of a hybrid-type IR algorithm for quantitative bronchial assessment on reduced-dose CT for patients with and without COPD and provided evidence of a significant correlation of WA% and the airway luminal volume percentage from the main bronchus to the peripheral bronchi (LV%) between standard- and reduced-dose CT [58]. Moreover, LV% agreement between standard-dose and reduced-dose CTs can improve AIDR 3D in comparison with FBP [59]. Therefore, AIDR 3D can be recommended for quantitative COPD evaluation on ADCT in routine clinical practice.

Few studies have been published on radiation dose reduction for the use of ADCT for the quantitative assessment of morphological evaluation of ILD. However, the utility of commercial or proprietary artificial intelligences (AIs) using machine-learning methods by Canon Medical Systems has been evaluated for the management of various lung diseases, such as ILD, or the evaluation of therapeutic treatments for coronavirus disease 2019 (COVID-19) pneumonia, which is caused by severe acute respiratory syndrome coronavirus 2 (SARS-CoV-2) [61,62,63] (Figure 2).

These studies were the first to demonstrate the potential of AI in the evaluation of disease severity and therapeutic effect or of functional changes due to treatment with an accuracy similar to that attained by board-certified radiologists [61,62,63]. Although further investigations are warranted, the evaluation of ADCT by AI has opened new areas for the application of pulmonary functional ADCT in not only ILD but also other diseases.

## 4. Pulmonary Perfusion Evaluation

Starting in the late 1960s, pulmonary perfusion was mainly assessed by means of nuclear medicine studies such as perfusion scanning, perfusion single-photon emission tomography (SPECT), or SPECT fused with CT (SPECT/CT). Since the late 1990s, pulmonary perfusion has also been academically or clinically assessed by using non-contrast-enhanced (non-CE) or contrast-enhanced (CE) perfusion MR imaging (non-CE- or CE-MRI) with a variety of techniques [64,65,66,67,68,69,70]. However, the application of each perfusion MR imaging procedure remains limited because of several technical issues such as pulse sequence design and optimization, certain procedures including contrast media concentration differentiation for quantitative or qualitative evaluation, somewhat lower temporal resolution than CT, and image analysis software for different clinical purposes [64,65,66,67,68,69,70]. As a result, pulmonary perfusion evaluation on CT has been performed since 2008 by means of three different methods: (i) dual-energy CT (DECT), (ii) subtraction CT (subtracting non-CE- from CE-CT images), and (iii) dynamic first-pass CE-perfusion CT.

### 4.1. Dual-Energy CT with ADCT System

Since the dual-source CT was installed for clinical use in 2008, DECT has been tested to assess its clinical utility for not only pulmonary vascular diseases but also other thoracic diseases [67,68,71,72,73,74,75,76,77]. This was followed, until recently, by the installation for testing and clinical use of rapid tube voltage (kVp) switching, dual-layer, or split-beam techniques for DECT. Table 2 shows a summary of the state-of-the-art DECT techniques provided by major vendors for routine clinical practice.

Although it has been suggested that DECT, similar to perfusion scan or SPECT, is useful for the evaluation of pulmonary perfusion in various pulmonary diseases [67,68,71,72,73,74,75,76,77], few major studies have been conducted on DECT-based assessment for the ADCT system. New clinical studies are, therefore, warranted in the near future to evaluate the clinical relevance of DECT for ADCT system for patients with various pulmonary diseases.

### 4.2. Subtraction ADCT

In contrast to DECT for the ADCT system, pulmonary perfusion evaluation for ADCT has been made possible by use of the subtraction technique with the appropriate software [78,79,80,81,82,83,84]. An in vitro study demonstrated that the contrast-to-noise ratio (CNR) of subtraction ADCT was superior to that of DECT by assessment with different iodine contrast media phantoms [79,80], while another in vivo study confirmed the superior clinical potential of subtraction ADCT in comparison with that of CE-CT pulmonary angiography (CE-CTPA) or DECT [81,82,83]. Moreover, lung subtraction iodine mapping by subtraction ADCT was shown to perform significantly better than CE-CTPA for patients with chronic thromboembolic pulmonary hypertension (CTEPH) [82]. In addition, in comparison with perfusion SPECT as the reference standard, lung subtraction iodine mapping with CE-CTPA showed promising results, with a sensitivity of 81.3% and a specificity of 78.9%, for the assessment of pulmonary perfusion in patients with acute pulmonary thromboembolism (PE) [83]. Moreover, subtraction ADCT was quantitatively and qualitatively directly compared with DECT for the assessment of lung nodule enhancement [84]. That study demonstrated that the mean nodule enhancement for subtraction ADCT was significantly higher than that for DECT. Lastly, the nodule enhancement on subtraction ADCT was judged more often to be “highly visible” than that on DECT by each of the observers engaged in that study. Subtraction ADCT was, therefore, considered to have better potential than DECT for iodine enhancement depiction in lung nodules [84]. Although subtraction ADCT is not as frequently used as DECT, this technique is potentially superior to DECT for the evaluation in routine clinical practice of differences in iodine contrast medium concentration through pixel-by-pixel analysis for not only thoracic but also other diseases when used with the appropriate software.

### 4.3. Dynamic First-Pass CE-Perfusion ADCT

The use of quantitatively analyzed dynamic first-pass CE-perfusion CT by means of electron-beam CT was first reported in 2000 [85]. However, after the introduction of MDCT, the use of electron-beam CT for this type of CT examination was changed to MDCT. This was followed by reports by some investigators, published between 2000 and 2010, that quantitative assessment of tumor or nodule perfusion assessment had potential for the diagnosis of pulmonary nodules or lung cancer, as well as for therapeutic effect assessment of lung cancer patients undergoing conservative therapy [85,86,87,88,89,90,91,92]. However, the limited scan range attainable with dynamic scanning in the same table position or the variety of perfusion data obtained at different time points and positions within the scan range due to the helical scan method remained major drawbacks of this technique until 2007 [87,88,89]. Since then, real dynamic first-pass CE-perfusion ADCT data in the form of isotropic volume data can be obtained by means of continuous dynamic scanning, allowing for the qualitative and quantitative evaluation of the perfusion of pulmonary nodules, lymph nodes, and tumors within a 160 mm area [5,12,14,15,16,17,18,19,20,21,22]. As a result, ADCT systems are now being used for both morphologic examinations and functional assessments, especially real first-pass perfusion evaluation, by means of the dynamic first-pass CE-perfusion ADCT technique using the appropriate mathematical models [5,12,14,15,16,17,18,19,20,21,22]. Table 3 lists the major clinical evidence with regard to dynamic first-pass CE-perfusion ADCT published during the last few decades.

For the diagnosis of pulmonary nodules, the diagnostic performance of dynamic first-pass CE-perfusion ADCT was equal to or significantly better than that of FDG-PET/CT or dynamic first-pass CE-perfusion MRI with a 1.5 T or 3 T MR system [15,21,22] (Figure 3).

Moreover, the diagnostic performance of dynamic first-pass CE-perfusion ADCT for lymph node metastasis was also shown to be equal to or significantly better than that of FDG-PET/CT for non-small-cell lung cancer (NSCLC) patients [20]. In addition, the dual-input maximum slope model was found to have better potential for accurate evaluation than the single-input maximum slope or Patlak plot methods in the aforementioned settings [19]. A comparison of the capability of response evaluation criteria to differentiate solid-tumor (RECIST) responders from RECIST non-responders in NSCLC patients treated with conservative therapy showed no significant differences in the sensitivity, specificity, and accuracy of dynamic first-pass CE-perfusion ADCT, MRI analyzed with the same dual-input maximum slope model, or FDG-PET/CT [22]. Furthermore, hybrid-type IR was shown to be more effective than FBP in terms of dose reduction for dynamic first-pass CE-perfusion ADCT while maintaining image quality and reducing measurement errors [16]. Dynamic first-pass CE-perfusion ADCT with an appropriate mathematical model as well as a reconstruction method, therefore, merits use as a pulmonary functional imaging method in routine clinical practice. Furthermore, Canon Medical systems is now developing and testing an appropriate protocol and proprietary software for the analysis of whole-lung dynamic first-pass CE-perfusion ADCT data and the creation of a whole-lung perfusion parameter map for a variety of different academic and clinical purposes, which will be made available for use in the near future.

### 4.4. Ventilation Assessment

Although the potential of Xe or krypton (Kr) to function as gas contrast media has been known since the 1960s [93,94,95], they were not academically or clinically used for pulmonary functional imaging until 2008. Oxygen-enhanced MRI and hyperpolarized noble gas MRI were reported to be useful for ventilation-based pulmonary functional imaging in the late 1990s [65,66,67,68,69,70]. For this reason, Xe- or Kr-enhanced ventilation CTs have mainly been used for pulmonary functional imaging for in vitro or in vivo studies after the clinical installation of DECT since 2008 [67,68,73,87,96,97,98,99,100,101,102,103,104,105,106,107,108,109,110,111]. Since that time, DECT has been mainly used by dual-source CT systems for patients with COPD or asthma [67,68,73,87,96,97,98,99,100,101,102,103,104,105,106]. However, no studies on the use of ADCT for Xe-enhanced DECT were published until 2023. In contrast to DECT, subtraction ADCT was demonstrated to be as effective as subtraction CT for the visualization of xenon enhancement for in vitro or in vivo studies and for pulmonary functional loss evaluation in comparison with the use of DECT or Kr-81m ventilation SPECT/CT for in vivo studies [107,108,109,110,111]. Moreover, these in vivo studies have demonstrated the potential of Xe-enhanced subtraction ADCT for regional ventilation evaluation or therapeutic effect assessment for smokers, COPD patients, asthmatics, or lung cancer patients [107,108,109,110,111] (Figure 4).

In addition, inspiratory and expiratory (inspiratory/expiratory) Xe-enhanced subtraction CT with hybrid-type IR has been found to be effective for the assessment of the regional ventilation changes in lung cancer patients due to smoking-related COPD or surgical treatment [110,111]. Although Xe-enhanced subtraction ADCT as well as DECT with dual-source CT are considered to be useful for pulmonary ventilation imaging, cold Xe or Kr has currently not been approved as contrast media for ventilation CT because they were only approved as contrast media for brain CT by the United States Food and Drug Administration, the Japan Pharmaceuticals and Medical Devices Agency, or regulatory authorities in other countries. It is, therefore, vital to obtain, in the near future, the regulatory approval of Xe and Kr as gas contrast media for pulmonary functional imaging, whether Xe-enhanced subtraction ADCT or DECT with dual-source CT are applied.

### 4.5. Biomechanical Evaluation

Pulmonary functional imaging has been recommended as having the potential to estimate lung compliance by tracking voxel motion at full inspiration (total lung capacity) and at full expiration (residual volume) by means of inspiratory/expiratory CT or MRI [67,68,69,70,112]. Moreover, lung compliance estimates derived from free-breathing and static-volume 4D CT and 4D MRI can be obtained by using deformable image registration [67,68,69,70,113,114,115,116,117,118,119,120,121,122,123]. Regarding the use of ADCT for the evaluation of lung biomechanics, the ACTIve study group has published in vitro and in vivo studies of the capability of dynamic 4D ADCT to evaluate regional lung biomechanics and pleural invasion in lung cancer patients as well as air trapping or air flow limitations [115,116,117,118,119,120,121,122,123]. Moreover, it is now possible to generate whole lung 4D ADCT images with a proprietary software provided by Canon Medical Systems and, in the near future, to begin to evaluate the efficacy of 4D ADCT as a new system for pulmonary functional imaging.

## 5. Conclusions

ADCT merits consideration as a useful tool for pulmonary functional imaging not only to evaluate pulmonary morphology but also, with various techniques, to directly evaluate pulmonary function. This system has been clinically tested as an upright CT system at our institution (Figure 5).

Moreover, it can be expected that some additional technical enhancements will be introduced in this decade on the basis of future studies using newly developed X-ray tubes, detectors, reconstruction methods, or other new technologies. When considering the future advancements of ADCT, faster rotation speed of the X-ray tube, higher isotropic resolution and larger image matrices for ultra- or super-high-resolution CT imaging, faster image processing, and computer-aided diagnosis or artificial intelligence tools are warranted to open new imaging methods for not only oncologic but also pulmonary functional imaging. Therefore, ADCT can be currently used as a standard pulmonary functional imaging tool in routine clinical practice but may soon lead to new pulmonary functional imaging functions.

## Figures and Tables

**Figure 1 diagnostics-13-02518-f001:**
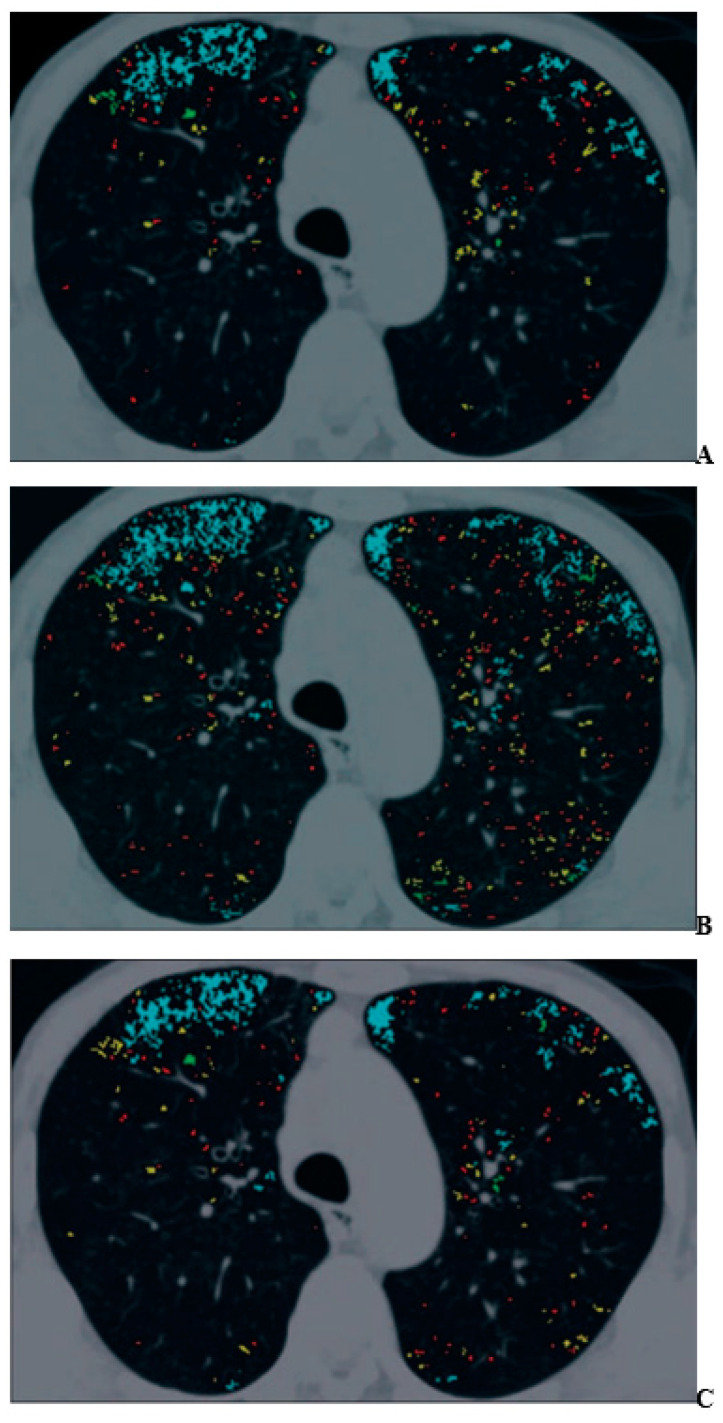
A 70-year-old man with mild pulmonary emphysema (permission from reference [44]). (**A**–**C**) Images show low-attenuation lung regions with standard-dose CT (**A**), low-dose CT without adaptive iterative dose reduction using 3D processing (**B**), and low-dose CT with adaptive iterative dose reduction using 3D processing (**C**). Color coding of low-attenuation lung regions is as follows: class 1, red; class 2, yellow; class 3, green; and class 4, cyan.

**Figure 2 diagnostics-13-02518-f002:**
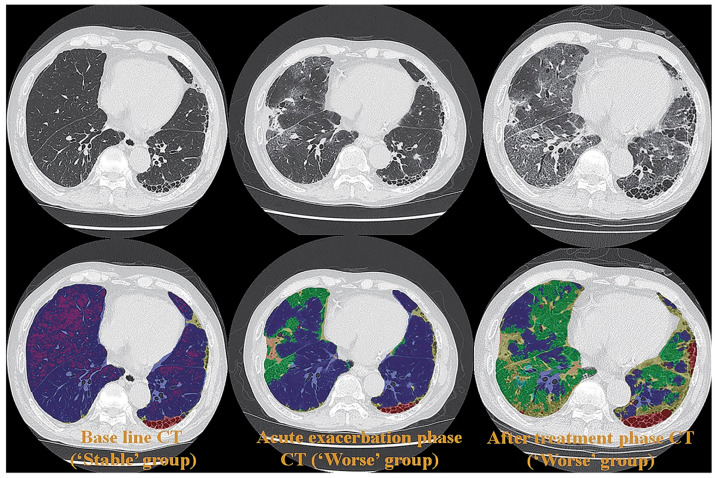
A 65-year-old female patient with progressive scleroderma (top row, L to R: thin-section CT scans at baseline, acute exacerbation phase, and after-treatment phase; bottom row, L to R: CT texture analysis by means of machine-learning-based software at baseline and the same two phases as for the top row) (permission from reference [63]). On machine-learning software, normal lung, consolidation, emphysema, GGO, honeycombing and reticulation are expressed as blue, beige, purple, green, red and green. A comparison of CT scans obtained at baseline (i.e., “Stable” group) and at the acute exacerbation phase (i.e., “Worse” group) shows an increase in the GGO and the consolidation area and a decrease in the normal lung area. Δ% normal lung, Δ% GGO, and Δ% consolidation were −16.9%, 13.2%, and 2.5%, respectively, while Δ disease severity score was 6. A comparison of CT scans obtained at the acute exacerbation phase (i.e., “Worse” group) and the after-treatment phase (i.e., “Worse” group) shows an increase in the GGO, reticulation, and honeycomb area and a decrease in the normal lung area. Δ% normal lung, Δ% GGO, Δ% reticulation, and Δ% honeycomb were −19.5%, 14.9%, 4.2%, and 0.2%, while Δ disease severity score was 15. CT, computed tomography; GGO, ground-glass opacity.

**Figure 3 diagnostics-13-02518-f003:**
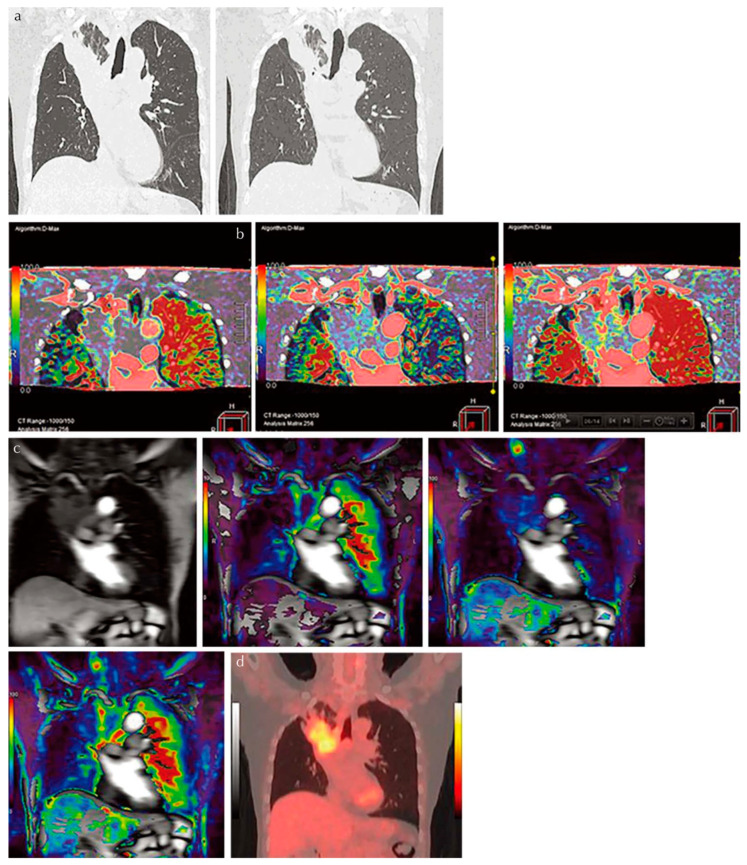
An 81-year-old male patient with squamous cell carcinoma treated with chemoradiotherapy and assessed as NC. Progression-free and overall survivals at 15 and 24 months (permission from reference [22]). (**a**) Thin-section MPR image derived from thin-section CT data (L to R: MPR images obtained pre- and post-treatment at lung window setting) show lung cancer in the right upper lobe. This case was assessed as NC according to response evaluation criteria for solid tumors (RECIST ver.1.1). (**b**) Perfusion maps derived from dynamic first-pass CE-perfusion area-detector CT assessed with the dual-input maximum slope method (L to R: pulmonary arterial perfusion, systemic arterial perfusion, and total perfusion maps) for the same targeted lesion. Pulmonary arterial perfusion, systemic arterial perfusion, and total perfusion were 13.6, 18.9, and 32.5 mL/100 mL/min, respectively. This case was assessed as a RECIST-based non-responder for systemic arterial and total perfusions and as true positive. (**c**) Source image and perfusion maps obtained with dynamic first-pass CE-perfusion MR imaging assessed with the dual-input maximum slope method (L to R: source image, pulmonary arterial perfusion, systemic arterial perfusion, and total perfusion maps) for the same targeted lesion. Pulmonary arterial perfusion, systemic arterial perfusion, and total perfusion were 9.2, 28.9, and 38.1 mL/100 mL/min, respectively. This case was also assessed as a RECIST-based non-responder for systemic arterial and total perfusions and as true positive. However, this case was evaluated as responder and as false positive based on pulmonary arterial perfusion findings. (**d**) PET/CT shows high uptake of 2-[fluorine-18]-fluoro-2-deoxy-d-glucose, and SUV_max_ was evaluated as 4.7. This case was evaluated as a RECIST-responder and assessed as false negative. PR, partial response; MPR, multiplanar reformatted; RECIST, Response Evaluation Criteria in Solid Tumors; CE, contrast-enhanced; SUV, standardized uptake value; PET, positron emission tomography.

**Figure 4 diagnostics-13-02518-f004:**
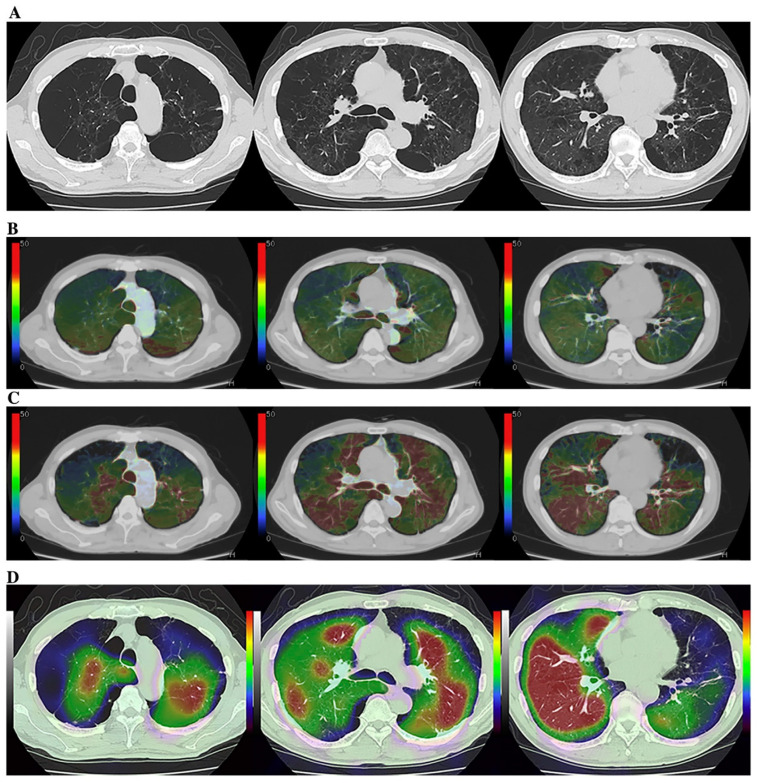
A 75-year-old male smoker with “Moderate COPD” (58 pack-years, FEV1/FVC%: 55%, %FEV1: 58%) (permission from reference [108]). (**A**) (L to R: cranial to caudal): Thin-section CT shows a heterogeneously distributed low attenuation area due to pulmonary emphysema. (**B**) (L to R: cranial to caudal): DECT shows heterogeneous xenon enhancement within the lung and areas of reduced xenon enhancements corresponding well to the distribution of the areas of low attenuation. Total ventilation defect score was 66, and functional lung volume was 65.8%. (**C**) (L to R: cranial to caudal): Subtraction CT shows heterogeneous xenon enhancement within the lung and areas of reduced xenon enhancement corresponding well to the distribution of the areas of low attenuation. Total ventilation defect score was 88, and functional lung volume was 53.7%. (**D**) (L to R: cranial to caudal): Co-registered Kr–81m ventilation SPECT/CT shows markedly heterogeneous uptakes within the lung. Regional uptakes correspond well to areas of low attenuation. Total ventilation defect score was 84, and functional lung volume was 55.8%. Regional uptakes of Kr–81m show better correspondence to xenon enhancement on subtraction CT than on DECT.

**Figure 5 diagnostics-13-02518-f005:**
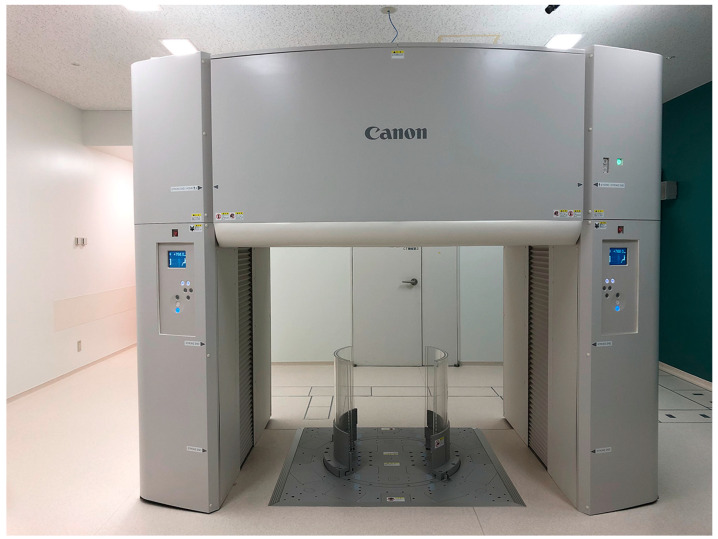
The first installed and clinically available upright CT with area-detector CT system in Fujita Health University Hospital.

**Table 1 diagnostics-13-02518-t001:** Hybrid-type iterative reconstruction, model-based iterative reconstruction, and deep learning reconstruction methods provided by major CT vendors.

Vendor	Reconstruction Methods		
	Hybrid-Type IR	Model-Based IR	DLR
Canon Medical Systems	Adaptive Iterative Dose Reduction 3D (AIDR 3D)	Forward Projected Model-Based Iterative Reconstruction Solution (FIRST)	Advanced intelligent Clear-IQ Engine (AiCE)
GE Healthcare	Adaptive Statistical Iterative Reconstruction (ASiR)	Veo	TrueFidelity
Philips Healthcare	4th-Generation Iterative Reconstruction (iDose^4^)	Iterative Model Reconstruction (IMR)	Precise Image
Siemens Healthineers	Iterative Reconstruction in Image Space (IRIS)	N/A	N/A
	Sinogram Affirmed Iterative Reconstruction (SAFIRE)		
	Advanced Modeled Iterative Reconstruction (ADMIRE)		

IR: iterative reconstruction, DLR: deep learning reconstruction.

**Table 2 diagnostics-13-02518-t002:** Dual-energy CT techniques currently available from major vendors.

Multienergy CT Technique	Dual Source	Split Beam	Rapid kVp Switching		Dual-Layer Detector
CT vendors	Siemens Healthineers		GE Healthcare	Canon Medical Systems	Philips Healthcare
Number of X-ray tubes	2	1	1	1	1
Scan time (sec/rotation)	0.25	0.28	0.28	0.275	0.27
FOV	Small in one X-ray tube	Full	Full	Full	Full
Z-axis coverage/rotation (mm/rot)	57.6–80	40	80	40–160	40–80
Automatic exposure control	Yes		No	Yes	Yes
Cross scattering	Yes		No	No	No
Filter	Yes		No	No	No
Registration	Slight temporal offset	Poor	Good	Good	Good
Spectral reconstruction method	Image		Projection and image	Projection and image	Projection and image
Tube current optimization for different energy bin	Yes	No	No	No	No
Spectral separation	Good	Limited	Good	Good	Limited

**Table 3 diagnostics-13-02518-t003:** Major clinical evidence for dynamic first-pass CE-perfusion ADCT.

Authors	Method	Target	Parameters	Cutoff Values (HU)	SE (%)	SP (%)	AC (%)
Ohno Y, et al. [5]	320-detector row CT	Pulmonary nodules	Perfusion (mL/100 mL/min) calculated with single-input maximum slope method	40.0	98 (42/43)	79 (26/33)	90 (68/76)
			Extraction fraction (mL/100 mL/min)	2.0	88 (38/43)	82 (27/33)	86 (65/76)
			Blood volume (mL/100 mL)	2.0	86 (37/43)	54 (18/33)	72 (55/76)
	FDG-PET/CT		SUV_max_	2.0	91 (39/43)	52 (17/33)	74 (56/76)
Ohno Y, et al. [12]	320-detector row CT	Pulmonary nodules	Total perfusion (mL/100 mL/min) calculated with dual-input maximum slope method	40	86.0 (49/57)	79.5 (31/39)	83.3 (80/96)
			Perfusion (mL/100 mL/min) calculated with single-input maximum slope method	20	64.9 (37/57)	69.2 (27/39)	66.7 (64/96)
	FDG-PET/CT		SUV_max_	2.5	63.2 (36/57)	56.4 (22/39)	60.4 (58/96)
Ohno Y, et al. [15]	320-detector row CT	Pulmonary nodules	Total perfusion (mL/100 mL/min) calculated with dual-input maximum slope method	29	92 (123/133)	71 (60/85)	84 (183/218)
			Nodule perfusion (mL/100 mL/min) calculated with single-input maximum slope method	10	91 (121/133)	28 (24/85)	67 (145/218)
	Dynamic first-pass CE-perfusion MRI for 1.5T system		Maximum relative enhancement	0.13	92 (123/133)	49 (42/85)	76 (165/218)
			Slope of enhancement	0.016	93 (124/133)	49 (42/85)	76 (166/218)
	FDG-PET/CT		SUV_max_	2	89 (119/133)	31 (26/85)	67 (145/218)
Ohno Y, et al. [20]	320-detector row CT	Lymph node metastasis in NSCLC analyzed per node	Total perfusion (mL/100 mL/min) calculated with slope of enhancement dual-input maximum slope method	58	54.2 (32/59)	89.8 (53/59)	72.0 (85/118)
			Systemic arterial perfusion (mL/100 mL/min) calculated with dual-input maximum slope method	4.1	98.3 (58/59)	56.4 (51/59)	92.4 (109/118)
			Permeability surface (mL/100 mL/min) assessed with Patlak plot method	8.7	50.8 (30/59)	94.9 (56/59)	72.9 (86/118)
			Distribution volume (mL/100 mL) assessed with Patlak plot method	0.37	84.7 (50/59)	44.1 (26/59)	64.34 (76/118)
	FDG-PET/CT		SUV_max_	2.9	74.6 (44/59)	91.5 (54/559)	83.1 (98/118)
Seki S, et al. [22]	320-detector row CT	Therapeutic outcome prediction for NSCLC	Total perfusion (mL/100 mL/min) calculated with dual-input maximum slope method	29.2	78.3 (18/23)	85 (17/20)	81.4 (35/43)
			Pulmonary arterial perfusion (mL/100 mL/min) calculated with dual-input maximum slope method	15.5	65.2 (15/23)	80 (16/20)	72.1 (31/43)
			Systemic arterial perfusion (mL/100 mL/min) calculated with dual-input maximum slope method	11	82.6 (19/23)	80 (16/20)	81.4 (35/43)
	Dynamic first-pass CE-perfusion MRI at 3T system		Total perfusion (mL/100 mL/min) calculated with dual-input maximum slope method	37.5	69.6 (16/23)	95 (19/20)	81.4 (35/43)
			Pulmonary arterial perfusion (mL/100 mL/min) calculated with dual-input maximum slope method	16.3	65.2 (15/23)	80 (16/20)	72.1 (35/43)
			Systemic arterial perfusion (mL/100 mL/min) calculated with dual-input maximum slope method	16.5	82.6 (19/23)	80 (16/20)	81.4 (35/43)
	FDG-PET/CT		SUV_max_	5.7	87.0 (20/23)	76.9 (14/20)	79.1(34/43)

SE: sensitivity, SP: specificity, AC: accuracy, NSCLC: non-small-cell lung cancer; number following author’s name corresponds to number in References.

## Data Availability

Not applicable.

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
