# Peer review of "Area-Detector Computed Tomography for Pulmonary Functional Imaging"

_diagnostics, 2023, doi:10.3390/diagnostics13152518_

Round 1

Reviewer 1 Report

Future trends to be included as a short comment

I suggest you the article MDCT: Technical Principles and Future Trends by Mathias Prokop

Author Response

Response to review comments for Manuscript ID (diagnostics-2463333)

Reply to Comments and Suggestions for Authors from Reviewer #1:

  1. Future trends to be included as a short comment. I suggest you the article MDCT: Technical Principles and Future Trends by Mathias Prokop. 

According to reviewer #1's comment #1, the suggested information is added in Page 15, line 378-383. 

Reply to Comments and Suggestions for Authors from Reviewer #2:

  1. Table 3, 4th row has a single line struck out (0.0016).

According to reviewer #1's comment #1, the statement is corrected in Table 3 in Page 9. 

  1. Overall very well-written review.

Thank you for your comments. 

  1. -Section 3: %LAA would correlate with PFT DLCO and WA% will correlate with FEV1/FVC or %FEV1 in spirometry. Maybe adding comments and references on this type of imaging to functional testing if available?

According to reviewer #2's comment #3, the suggested statement is added in Page 4, line 142-146. 

  1. -Section 4.4 can specify why cold Xe or Kr imaging has not been approved (discussion of one current gap?).

According to reviewer #2's comment #4, the statement is modified in Page 13, line 347-Page 14, line 352. 

  1. -Section 4.4 would need to be tightened up a little better. Ending it here with the Canon system that is about to be evaluated seems not aligned with how Section 4.3 ended by stating that he regulatory bodies have not approved Xe or Kr studies. Try to keep the enthusiasm or conclusion similar in these two sections to remove the appearance of bias here.

According to reviewer #2's comment #5, the statement is modified in Page 14, line 352-354. 

  1. -Section 4.5 can be expanded with more reference and explanation as to why this biomechanics evaluation is clinically relevant (meaning correlation to other clinical markers). Similar in ideas as comments on other section above.

Thank you for your comments.  Although we agree to reviewer #2’s comment #6, there were limited references for biomechanics evaluation on ADCT.  Reference #116-126 had been stated in this version.  Therefore, additional comments might not be added in this time point. 

Reviewer 2 Report

This review manuscript focuses on ADCT's new reconstruction methods, pulmonary ventilation, and perfusion imaging.

Table 3, 4th row has a single line struck out (0.0016).

Overall very well-written review.

-Section 3: %LAA would correlate with PFT DLCO and WA% will correlate with FEV1/FVC or %FEV1 in spirometry.  Maybe adding comments and references on this type of imaging to functional testing if available?

-Section 4.4 can specify why cold Xe or Kr imaging has not been approved (discussion of one current gap?).

-Section 4.4 would need to be tightened up a little better.  Ending it here with the Canon system that is about to be evaluated seems not aligned with how Section 4.3 ended by stating that the regulatory bodies have not approved Xe or Kr studies.  Try to keep the enthusiasm or conclusion similar in these two sections to remove the appearance of bias here.

-Section 4.5 can be expanded with more reference and explanation as to why this biomechanics evaluation is clinically relevant (meaning correlation to other clinical markers).  Similar in ideas as comments on other section above.

Author Response

(The authors gave the same response as above.)
